# Pathogenicity and Relative Abundance of *Dickeya* and *Pectobacterium* Species in Switzerland: An Epidemiological Dichotomy

**DOI:** 10.3390/microorganisms9112270

**Published:** 2021-10-31

**Authors:** Patrice de Werra, Christophe Debonneville, Isabelle Kellenberger, Brice Dupuis

**Affiliations:** 1Plants and Plant Products, Varieties and Production Techniques, Agroscope, 1260 Nyon, Switzerland; patrice.dewerra@agroscope.admin.ch; 2Plant Protection, Virology, Bacteriology and Phytoplasmology, Agroscope, 1260 Nyon, Switzerland; christophe.debonneville@agroscope.admin.ch (C.D.); isabelle.kellenberger@agroscope.admin.ch (I.K.)

**Keywords:** potato, blackleg, Soft Rot Pectobacteriaceae, pathogenicity

## Abstract

*Pectobacterium* and *Dickeya* species are the causal agents of blackleg and soft rot diseases in potatoes. The main pathogenic species identified so far on potatoes are *Dickeya dianthicola*, *Dickeya solani*, *Pectobacterium atrosepticum*, *Pectobacterium brasiliense*, *Pectobacterium carotovorum*, and *Pectobacterium parmentieri*. Ten years ago, the most prevalent Soft Rot Pectobacteriaceae in Europe were the *Dickeya* species, *P. atrosepticum* and *P. carotovorum*, with some variations among countries. Since then, a drastic increase in the abundance of *P. brasiliense* has been observed in most European countries. This shift is difficult to explain without comparing the pathogenicity of all *Dickeya* and *Pectobacterium* species. The pathogenicity of all the above-mentioned bacterial species was assessed in field trials and in vitro tuber slice trials in Switzerland. Two isolates of each species were inoculated by soaking tubers of cv. Desiree in a suspension of 10^5^ CFU/mL, before planting in the field. For all trials, the *Dickeya* species were the most virulent ones, but long-term strain surveys performed in Switzerland indicate that *P. brasiliense* is currently the most frequent species detected. Our results show that the pathogenicity of the species is not the main factor explaining the high prevalence of *P. brasiliense* and *P. parmentieri* in the Swiss potato fields.

## 1. Introduction

Blackleg and tuber soft rot caused by Soft Rot Pectobacteriaceae (SRP) are among the most important diseases of potato (*Solanum tuberosum*). The overall annual loss for the European potato sector due to SRP is estimated at EUR 46 million [1].

In Europe, the six main SRP species are the following: *Pectobacterium carotovorum*, *Pectobacterium brasiliense*, *Pectobacterium parmentieri* (formerly named *Pectobacterium wasabiae*), *Pectobacterium atrosepticum*, *Dickeya dianthicola* and *Dickeya solani* [2,3,4,5]. The relative abundance of each bacterial species has changed over the years and spatially. The population composition of species has evolved differently from one region to the other. To illustrate these changes, Scotland, the Netherlands and Switzerland are used as examples. In Scotland, the most prevalent species in the most recent surveys is *P. atrosepticum* [4]. Scotland is still free from *Dickeya* species and has a ‘nil’ tolerance for *Dickeya* spp. [6]. In the Netherlands, *P. atrosepticum* was the most prevalent species until the early 2000s, but its prevalence subsequently decreased in favor of the two aforementioned *Dickeya* species [4,7]. Ten years later, *P. brasiliense* and *P. parmentieri* were also reported in the Netherlands, and the prevalence of *P. brasiliense* has rapidly increased [3]. In Switzerland, *Dickeya* species were prevalent since the beginning of the first SRP surveys undertaken in the 1980s [8,9]. In 2013, *P. brasiliense* was detected in about 10% of the blackleg samples collected in the field; one year later, this species was identified in about 90% of the blackleg samples collected in the country [10]. This remarkable switch in population was confirmed in the following years, with 80%, 57% and 58% of relative abundance respectively in 2014, 2015 and 2016 [11]. This switch cannot only be explained by the arrival of *P. brasiliense* in the country, as identification of historical samples has proven that this SRP species was already present in Switzerland in 1988. *P. brasiliense* was formerly identified as *P. carotovorum* by ELISA testing [10].

Many hypotheses have been offered to explain the changes in SRP populations. Climate change and, more specifically, warmer growing conditions were proposed as an explanation for the increase of the prevalence of *Dickeya* species in Europe [4,12,13]. Indeed, *Dickeya* species are known to have a tropical origin and are particularly virulent in hot climate conditions [14,15]. This is consistent with the fact that *Dickeya* species are still absent in Scotland, where the growing conditions are colder [12], while they were present in Switzerland long before the Netherlands [10,16], where the growing conditions are colder than those in Switzerland. A higher pathogenicity, defined as the qualitative ability of a pathogen to cause disease, and virulence, defined as the quantitative degree of damage caused to the host [17], was proposed as an explanation for the increase in the prevalence observed with *P. brasiliense* [3]. However, we know that this species was already present in Switzerland long before the increase in its population observed from early 2010 onwards [10]. Nevertheless, we cannot exclude the possibility that the former isolated strains were less aggressive than the new ones, due to some genetic variability, as suggested previously [10]. To clarify this situation, pathogenicity trials were undertaken in the Netherlands with both *Pectobacterium* and *Dickeya* species under field conditions for two years [3]. In these trials, the isolates of *P. brasiliense* showed the highest incidence in the field (>80% of disease incidence), closely followed by isolates of *P. atrosepticum* (between 70% and 80% disease incidence), while *D. solani* isolates showed relatively few symptoms (between 5% and 30% disease incidence), and almost no symptoms were observed with *D. dianthicola* isolates (<5% disease incidence). The discrepancy between the latter and the results of field trials performed for two years in Switzerland is surprising [18]. In these trials, an isolate of *D. dianthicola* (88/23) was shown to be the most pathogenic, compared to three isolates of *D. solani* and another *D. dianthicola* tested. High pathogenicity of some *D. dianthicola* isolates, compared to *D. solani* isolates was also observed in other field trials performed in the Netherlands, Finland and Switzerland [19,20,21]. This demonstrates that the pathogenicity of *D. dianthicola* varies according to the isolate tested, while the pathogenicity of *D. solani* appears to be relatively homogenous. Interestingly, the strains of *D. solani* are known to be genetically more clonal than the strains of *D. dianthicola* present in Europe [22,23,24,25]. The variability of the pathogenicity of *D. dianthicola* isolates was also observed in greenhouse trials studying the development of blackleg symptoms [19,26], as well as in trials with the inoculation of tubers examining the development of soft rot symptoms [22,27,28,29]. Therefore, it would be of interest to compare the pathogenicity of highly virulent *D. dianthicola* isolates (such as 88/23) with isolates of *P. brasiliense* to define whether the latter species is more pathogenic than virulent isolates of *D. dianthicola*. In this case, it would explain the rapid increase in the prevalence of *P. brasiliense* observed in some European countries in recent years. In addition, the pathogenicity of *P. brasiliense* has not yet been compared to the other species in soft rot trials. Exploring the pathogenicity and virulence of *P. brasiliense* for soft rot disease compared to the other SRP species would provide valuable information for a better understanding of its epidemiology.

In this research, we compared the virulence of *D. dianthicola*, *D. solani*, *P. atrosepticum*, *P. brasiliense*, *P. carotovorum*, and *P. parmentieri* in a field trial over two years, and a trial with inoculated tuber slices. We also carried on with the Swiss long-term survey of pathogenic strains of *Dickeya* and *Pectobacterium* to document the evolution of the relative abundance of those bacteria in Switzerland. The results are discussed for a better understanding of the evolution of the populations of SRP species observed in the different pedoclimatic regions of Europe.

## 2. Materials and Methods

### 2.1. Bacterial Strains and Species Identification

The bacterial isolates used to inoculate the tubers were two strains of *Pectobacterium carotovorum* (Imp16/645-4 and 81/N65), two strains of *Pectobacterium brasiliense* (88/157-2 and CH14/110-4), *Pectobacterium parmentieri* (93/39-2 and 2013-PP88), *Pectobacterium atrosepticum* (08/023 and 16/021-3-1), *Dickeya dianthicola* (13/089-3 and 88/23) and *Dickeya solani* (07/044 and 14/188-1) (Table 1). All the strains came from the Agroscope collection in Changins (VD, Switzerland). The strains were randomly selected from the collection based on the following criteria: (i) they were isolated in Switzerland from a symptomatic blackleg diseased plant; (ii) when possible, a “historical” and a recently isolated strain were chosen for each species (see Table 1). Except for *Dickeya dianthicola* 88/23, known as being highly virulent from previous studies [18], the strains were not selected based on any virulence criteria to avoid bias in selecting particular aggressive strains.

For the identification of bacterial strains from the living collection, genomic bacterial DNA was extracted using the Wizard genomic DNA purification kit (Promega). Downstream specific PCR amplification was performed on 1 µL of DNA extraction. Each reaction mixture (25 µL) contained 1 × GoTaq green buffer (Promega), 3 mM MgCl_2_, 0.2 mM dNTPs (Fisher Scientific), 1 µM forward primer, 1 µM reverse primer, and 1 U of GoTaq polymerase (Promega). *Dickeya* species were detected using the primers ADE1 and ADE2 [30]; *Pectobacterium atrosepticum* was detected, using the primers ECA1f and ECA2r [31]; *Pectobacterium brasiliense* was detected, using the primers BR1f and L1r [32]; *Pectobacterium carotovorum* was detected, using the primers EXPCCF and EXPCCR [33]; and *Pectobacterium* (*wasabiae*) *parmentieri* was detected, using the primers PW7011F and PW7011R [34]. The cycling conditions were set as follows: 5 min at 94 °C for 35 cycles (45 s at 94 °C, 45 s at 62 °C, 1.5 min at 72 °C) followed by a final extension of 7 min at 72 °C for *P. atrosepticum*, *P. carotovorum* and *P. brasiliense*; 5 min at 94 °C for 35 cycles (1 min at 94 °C, 30 s at 67 °C, 1 min at 72 °C) followed by a final extension of 7 min at 72 °C for *Dickeya* spp. and *P. parmentieri*. Moreover, to differentiate both species of *Dickeya*, supplementary PCR was performed, using specific primers sets: DIA-A and DIA-C for *D. dianthicola* and SOL-C and SOL-D for *D. solani* [35].

### 2.2. Inoculation of Tubers for Field Trials

Tuber inoculations were carried out as described in Dubois Gill et al. [36]. The main steps of the procedure are as follows: soaking the potato tubers in water for 2 h, storage for 22 h at 25 °C and 80% RH to allow the lenticels to open, soaking in a bacterial suspension of 10^5^ CFU/mL for 12 h and, finally, drying for 12 h. The tubers used as controls were soaked in tap water instead of the bacterial suspension. In Switzerland, tap water is treated, and therefore, it is expected to be free from potato pathogenic bacteria.

### 2.3. Field Trials

Field trials were performed over two years. In the first year (2018), the trial was conducted in Goumoens-la-Ville (VD, Switzerland) and the second year (2020) in Reckenholz (ZH, Switzerland). The first year, the trial was planted on 16 April 2018, and the second trial started on 8 April 2020. Plots were planted in a loamy soil in both years (24% clay, 39% loam, 37% sand in 2018, and 21% clay, 31% loam, 48% sand in 2020), according to a completely randomized block design, with four replications. Each plot was planted with two rows of 50 plants of cv. Desiree inoculated with one of the 12 tested bacterial isolates and surrounded by two rows of cv. Challenger acting as a buffer zone. There were two uninoculated controls in each block. The inter-row spacing was 75 cm, and within a row, tubers were planted every 33 cm. The experimental field was managed following standard cultural practices without irrigation and 120 kg N/ha was added at planting (liquid nitrogen formulation). Potatoes were grown as a ware potato crop following a rotation with at least five years’ break between two potato crops. The number of plants presenting rotting stems was scored once a week. Meteorological data were retrieved from a weather station located less than 200 m from the experimental fields. Tubers were harvested about 3 weeks after haulm destruction. Yield was assessed for each replicate.

### 2.4. Tuber Slice Trial

The protocol used to perform this trial is an adaptation of the protocol proposed by Haynes et al. [37]. Tubers of cv. Desiree were firstly surface sterilized by soaking in 70% ethanol, and then briefly passed through the flame of a Bunsen burner. A slice, about 5 mm thick, was cut from the center of the tuber, and then placed in a Petri dish containing 1 mL of sterile water. A 1 cm^2^ filter paper was placed in the center of the slice. A first weighing was then carried out in order to determine the initial weight. Then, 100 μL of bacterial suspension (10^7^ CFU/mL in phosphate buffered saline (PBS)) was deposited on the filter paper. The box was sealed with Parafilm to limit gas exchange and incubated in a heat chamber at 27 °C for 48 h. After incubation, the rot caused by the bacteria was removed. A second weighing was carried out to obtain the final weight and calculate the weight loss. This weight loss corresponds to the proportion of the slice degraded by the bacteria. Each isolate was tested on 20 slices, and 20 control slices were used for the entire experiment. On the latter, PBS was applied instead of the bacterial suspension. Each slice was taken from a separate tuber.

### 2.5. Long-Term Strain Survey

Every year since 1986, during the growing season, Swiss potato growers and field inspectors have sent diseased-blackleg samples to Agroscope for analysis. From 2012 to 2016, supplementary samples were analyzed at the HAFL (School of Agricultural, Forest and Food Sciences, Bern University of Applied Sciences), using the same procedure. Before 2010, all of the samples were analyzed with an ELISA procedure (details not described). From 2010 to 2016, ELISA procedure was gradually disused, and the stems of the samples from the Swiss potato fields were analyzed, using the following standardized PCR-based approach: the plant stems were cut in half, plant tissue (50–100 mg) was removed from the interface of healthy and symptomatic parts and then placed in a 2 mL Eppendorf tube containing 1.5 mL of 1 × PBS. The tubes were gently agitated for 20 min at room temperature. Then, 2 µL of the resulting suspension was used for strain identification. The PCR-based methods used are the same as the ones described above. For the purpose of the Swiss long-term survey, the *Dickeya* species were not characterized, and only the genus was determined.

### 2.6. Statistical Analysis

For the field trials, determination of statistical differences was based on comparison of the disease indices (field incidence) at the last observation, where the maximum incidence is reached. First, the initial conditions of validation were tested to determine which approach, parametric or non-parametric, should be used. If conditions of normal residues distribution and homoscedasticity were fulfilled, a parametric approach based on an ANOVA followed by a pairwise multiple comparison (ad hoc test of Duncan’s method) was performed. Otherwise, a non-parametric approach based on a Kruskal–Wallis test was used. The analyses were performed, using the Excel Add-in XLSTAT software program [38].

## 3. Results

### 3.1. Field Trials

In the first field trial planted in early spring 2018 with artificially inoculated tubers, disease progression was monitored weekly until haulm destruction. In this trial, a regular progression of the disease was observed with both *D. solani* strains, both *D. dianthicola* strains and one *P. brasiliense* strain (Figure 1a). At the last observation, the incidence of blackleg symptoms for these strains was significantly higher than the water control in which some diseased plants were also observed. Both strains of *D. dianthicola* as well as both strains of *D. solani* presented similar disease progression curves. For *P. brasiliense*, the blackleg incidence of one strain differed from the water control, while the second strain did not significantly differ from the control. At the species level, *D. solani* and *D. dianthicola* were the two species that differed significantly from the water control (Figure 1b). No significant differences in yield were observed in the 2018 field assay (Figure 2).

A repetition of the 2018 field trial was planted in 2020 at Zürich Reckenholz, in a similar loamy soil. Tubers were artificially inoculated using the same protocol, with the same bacterial strains. *D. solani* and *D. dianthicola* presented the highest disease incidence, whereas the *Pectobacterium* species were in the same range as the water control. Due to considerable variability among the replicates, the conditions were not fulfilled to perform parametric analyses (ANOVA), so a non-parametric approach (Kruskal–Wallis test) was chosen. At the strain level (Figure 3a), only both of the *D. solani* strains differed from the water control. Differences between the two *D. solani* isolates were negligible, as for the *D. dianthicola* isolates. Considering the average of each species, the blackleg incidence for *D. solani* and *D. dianthicola* differed from the water control (Figure 3b).

Significant differences in yield between the different strains used in 2020 were detected, which was not the case of the 2018 field trial (Figure 4). A significant reduction of the yield was measured for species and strains of the genus *Dickeya*. The yields of the water control were not exactly the same between 2018 (about 300 kg/a) and 2020 (about 400 kg/a), pointing to the fact that yield is also strongly year dependent (climatic conditions, soil, fertilizer supply, haulm destruction). For both field trials, metrological data (daily average temperature and precipitation) are presented in Figure 5.

When combined, the results of the years 2018 and 2020 show that *D. solani* and *D. dianthicola* were the species responsible for the highest blackleg incidence in the field. The other tested species (*P. brasiliense*, *P. parmentieri*, *P. atrosepticum* and *P. carotovorum*) did not result in significant disease incidence, compared to the water control.

### 3.2. Tuber Slice Trial

The last assay presents the results of the in vitro potato tuber slices trial in 2020. In this assay, tuber slices of the variety Desiree were inoculated with the tested bacterial strains, and the weight of the rotting potato flesh was measured. The large number of replicates allowed a parametric approach for the statistical analyses. All the strains generated more macerated tissue than the water control, and all strains were able to induce rotting of the tuber slices (Figure 6a). No macerated tissues were obtained from tuber slices of the negative control. Looking at the overall soft rot incidence per species (Figure 6b), we observed the highest soft rot incidence for *D. solani*, and then for *P. carotovorum*, in the same range as *D. dianthicola*. Three species resulted in less than 10 g of rotting potato tissue, namely, *P. brasiliense*, *P. atrosepticum*, and *P. parmentieri*. For *P. brasiliense* and *P. parmentieri*, a large variability in soft rot incidence among the strains was observed, with one strain above 10 g of macerated tissue and the other below this limit (Figure 6a).

### 3.3. Strain Survey

In addition to in vitro and field trials, results up to 2021 of the Swiss long-term survey of SRP are presented in Figure 7, as a complement to a previous publication [11]. This presents the relative abundance of bacterial species over years from naturally diseased plant samples of growers from all around Switzerland. The main change to be observed on this chart is the shift in 2012 between the *Dickeya* spp. (regardless of the species, *dianthicola* or *solani*) and the new prevalent species *P. brasiliense*, and *P. parmentieri* to a lesser extent [11]. Parallel to this shift, *P. atrosepticum*, previously an important pathogen in Switzerland [9], drastically decreased in importance, if not completely disappeared from the survey. Onwards from 2018, the populations seem to be relatively stable, with more than 60% of *P. brasiliense*, less than 40% of *P. parmentieri*, and less than 20% of *Dickeya* spp. In 2021, we note that no *Dickeya* spp. was detected in the samples from the field.

As diseased plant samples for the strain survey come from all over Switzerland, it is difficult to provide corresponding weather data that are representative for all of them. Nevertheless, as most of the seed potatoes grown in Switzerland are grown on the “Swiss plateau” between the Alps and the Jura mountains, a table presenting the weather data collected in Düdingen, a city located in the center of the potato growing area, is provided in the Appendix A.

## 4. Discussion

The results presented in this study attempted to rank the pathogenicity of different SRP strains, using in vitro slices and field trials. Basically, *Dickeya* spp. were more virulent than *Pectobacterium* spp., as shown by trends in results from in vitro slices assays, and as significantly established for both field trials.

Results showed that field incidence could have, but not necessarily, an impact on the weight at harvest. It must be noted that losses due to soft rot disease after a long storage period were not tested in this experiment. Even if all strains tested are pathogenic and thus able to induce tuber soft rot in vitro, high field incidence was not necessarily followed by a yield reduction. Therefore, in vitro assays, even though these are informative to test pathogenicity, do not replace field experiments and are inappropriate to test the virulence of strains in a plant–pathogen system. Plant–pathogen interactions are much more complex in natural outdoor conditions (field) compared to an in vitro potato slice or whole tuber assay, mainly due to the influence of the environment. This underlines the high importance of field assays to establish a ranking of pathogenicity of different strains, even though they are time consuming and fastidious.

Disease field incidence is strongly related on isolate aggressiveness, cultivar susceptibility, and yearly geo-climatic conditions. It is, therefore, almost impossible to draw general conclusions and multi-purpose ranking pathogenicity of a panel of strains from a few local field assays. The pathogenic response of a panel in a given place can be completely different in another one; several worldwide studies support this statement [3,4,14,39,40,41].

Isolates from *D. dianthicola* and *D. solani* do have a very similar field incidence pattern, whereas isolates from *P. brasiliense* present different incidence values (mainly in 2018). Another study [4] concluded a higher virulence for *D. solani* isolates compared to *D. dianthicola*. Regarding these two *Dickeya* species, we observed contrasting results from one year to the other.

Differences in the virulence within a species could have been obtained due to our random selection of strains. These picking probabilities cannot be excluded. To have a more precise view of the variability of the virulence of all species, it would have been better to test more strains in our trials. Nevertheless, as we were constrained by the size of the field trial, we preferred to test all *Dickeya* and *Pectobacterium* species in the same trial, instead of testing more strains but fewer species. Considering the genus, it is, on the other hand, very unlikely that only virulent *Dickeya* isolates were randomly picked, and in the same way, only non-virulent *Pectobacterium* isolates chosen.

This study also presents the last three years of data from a long-term survey of the relative abundance of *Dickeya* and *Pectobacterium* species in Switzerland (Figure 7). These data are in line with the data collected since 2014 with a majority of *P. brasiliense* and a low relative abundance of *Dickeya* spp., while *Dickeya* species were the most frequently detected in Switzerland before 2014. For the first time in this historical survey, no *Dickeya* strains were detected in the field samples in 2021. This can be explained by the average low temperatures of the year. The average daily temperature from April to July in the center of the seed potato growing area (Düdingen, FR) was 13.1 °C in 2021, 1.3 °C lower than the average of the 10 previous years (Appendix A data retrieved from www.agrometeo.ch, accessed on 10 September 2021). *Dickeya* species are known to be more virulent under higher temperature regimes [14,15], which could explain why its relative abundance was null in 2021, compared to previous years.

Since 2013, an increase in disease outbreaks of *P. brasiliense* has been observed, followed by *P. parmentieri*, in Switzerland [42] as well as in other countries [3,43]. These two *Pectobacterium* species were predominantly found in the yearly survey up to now. The results of this experiment showed that this high abundance cannot be explained by a higher pathogenicity and virulence of *P. brasiliense* or *P. parmentieri*. At equal concentrations of inoculum, isolates of *D. dianthicola* and *D. solani* are much more aggressive and able to cause marked damages in field, as shown in this study.

As the high relative abundance of the *Pectobacterium* species (mainly *P. brasiliense*) in Switzerland cannot be attributed to its high virulence in the field, other factors must be involved to maintain this large frequency of distribution. It is important to emphasize that pathogenicity and virulence of the tested strains are negatively correlated to their relative abundance. In the actual Swiss situation, high virulence obviously does not lead to a higher abundance (Figure 7).

We propose three alternative arguments to explain this epidemiological dichotomy and consequently the shift of species recently observed in several European countries.

Firstly, the aggressiveness of the *Dickeya* strains present in the environment could have decreased over time. A similar hypothesis is proposed for recently isolated strains of *Phytophthora infestans*, which tend to be less aggressive than the earlier ones [44]. This observation is not, however, intuitive, as it would be expected that virulence will increase over time and serve the pathogen to settle itself in the potato plants population. However, a decrease of pathogenicity to the so-called optimal virulence could also allow the pathogen to maintain itself in the plant population [17]. Shortly described, virulence, considered to be parasite fitness, rests on two major components: within-host multiplication and between-host transmission. As high virulence results in higher host mortality, a negative correlation is expected with between-host transmission. Optimal virulence is, thus, determined by a tradeoff between within-host multiplication and between-host transmission: the trade-off hypothesis. With *Pectobacterium* species being less virulent than the *Dickeya* ones, a better transmission of bacterial strains to daughter tubers is assumed. This could explain the large abundance of *P. brasiliense* in the Swiss survey.

Secondly, the results obtained must be placed in the perspective of the method of inoculation used for our trials. For the field trials, tubers were inoculated through the lenticels. Therefore, the expression of blackleg symptoms observed in these experiments does not reflect the pathogenicity of *Dickeya* and *Pectobacterium* species for inoculum present in the vascular system of the mother tuber or taken up from the soil by the roots of the growing plants [45]. Nevertheless, previous field experiments performed in Switzerland with three strains of *Dickeya dianthicola* and three strains of *Dickeya solani* revealed that most pathogenic strains for plants inoculated through the lenticels were also the most pathogenic for plants inoculated through the vascular system of the mother tuber [46]. This implies that the method used to inoculate the plants is probably not a determining factor regulating the intensity of the symptoms induced by the strains that were tested in this research study.

Finally, the incidence of the *Dickeya* spp.–induced seed-borne blackleg disease in Switzerland may have decreased due to the fact that these strains show visible symptoms in the field that are easy to spot and thus, are easy to remove from the field by roguing. Therefore, the many efforts made by the Swiss seed potato sector to improve the quality of the seed lots regarding the infections by *Dickeya* species has probably paved the way for the increase of the prevalence of other bacterial species, such as *P. brasiliense* and *P. parmentieri*. Having said that, it is surprising that the shift in populations in Switzerland occurred over a very short interval of one or two years (Figure 7).

In summary, complex interactions between the potato varieties, the *Dickeya* and *Pectobacterium* strains present, and the environment features may be responsible for the recent evolution of the pectinolytic bacteria populations observed in certain European countries, such as Switzerland.

## 5. Conclusion

One should consider that the aggressiveness or virulence of a pathogen is positively correlated to its abundance in potato fields. This statement is probably true for many plant pathogens. The results presented in this study challenge this conventional wisdom, as the most virulent species identified in our field trials (*Dickeya* spp.) are not precisely the most prevalent in the potato fields in Switzerland (*Pectobacterium* spp.). However, this observed dichotomy between aggressiveness and prevalence support the idea that being too aggressive does not help pathogen fitness, as seed lots presenting a high blackleg rate are more likely to be excluded from multiplication, which is not favorable for the spread of aggressive strains. By contrast, the low virulence of *P. brasiliense* and *P. parmentieri* in the field trials performed in this study could contribute to explaining the epidemiological success of these species by reaching high prevalence under the Swiss field conditions.

## Figures and Tables

**Figure 1 microorganisms-09-02270-f001:**
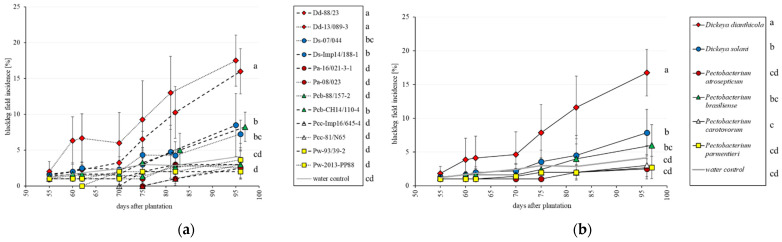
Timelines of the observed blackleg field incidence for the different 12 strains (**a**) and the different 6 species (**b**), average of two strains of similar color, in the field assay for 2018. Each single value of the timeline is the mean of four repetitions of 100 observed plants, with standard deviation bars. Different letters indicate statistical differences according to pairwise multiple comparison (ANOVA based) of the mean incidence value between the strains at the last observation. In the chart, letters are grouped together. Details statistical differences between strains or species are given right to the legends. Negative control with water is given in gray. The assay was repeated in 2020.

**Figure 2 microorganisms-09-02270-f002:**
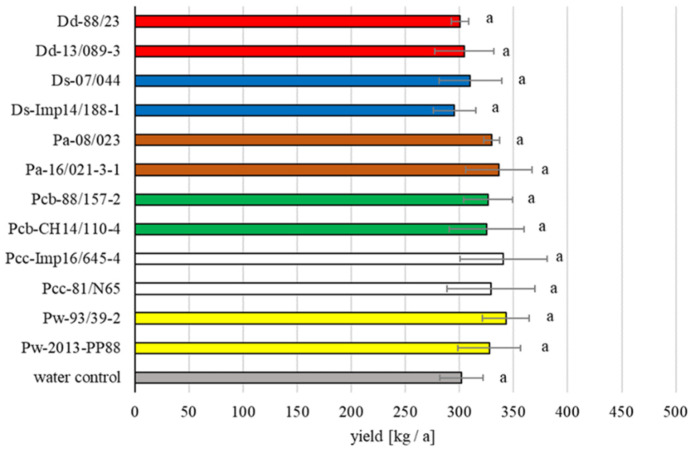
Yield in kg per are of the trials using potatoes infected with 12 bacterial strains in the field assay for 2018. Mean value of four repetitions for each strain, with standard deviation bars. No statistical difference was measured between the strains or the species. The assay was repeated in 2020.

**Figure 3 microorganisms-09-02270-f003:**
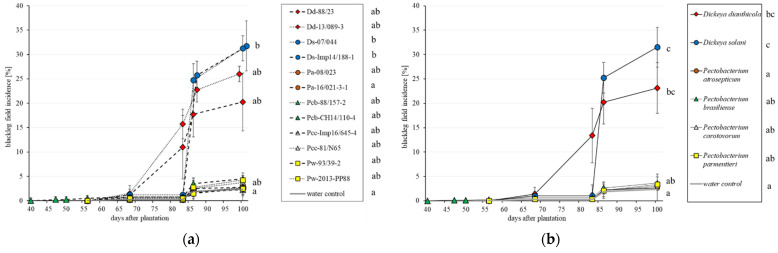
Timelines of the observed blackleg field incidence for the different 12 strains (**a**) and the different 6 species (**b**), average of two strains of similar color, in the field assay for 2020. Each single value of the timeline is the mean of four repetitions of 100 observed plants, with standard deviation bars. Different letters indicate statistical differences according to Kruskal-Wallis test (non-parametric-ANOVA based) of the mean incidence value between the strains at the last observation. In the chart, letters are grouped together. Details statistical differences between stains or species are given right to the legends. This is a repetition of the 2018 assay.

**Figure 4 microorganisms-09-02270-f004:**
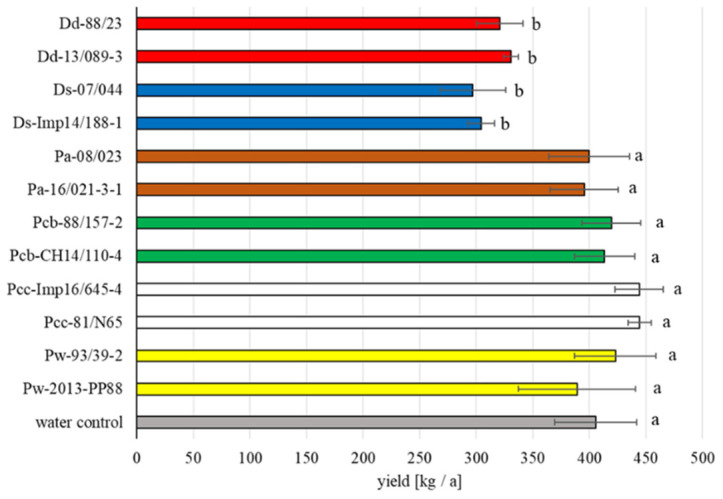
Yield in kg per are of the trials using potatoes infected with 12 bacterial strains in the field assay for 2020. Mean value of four repetitions for each strain, with standard deviation bars. Different letters indicate statistical differences according to pairwise multiple comparison (ANOVA based) of the mean incidence value between the strains. This is a repetition of the assay 2018.

**Figure 5 microorganisms-09-02270-f005:**
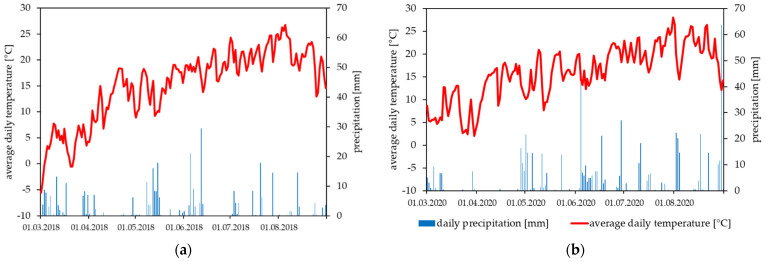
Meteorological data for the field trials Goumoens-la-Ville (VD, Switzerland) 2018 (**a**) and Reckenholz (ZH, Switzerland) 2020 (**b**). Average daily temperature (°C) and daily precipitation (mm) are given over a period, including the potato cultivation period.

**Figure 6 microorganisms-09-02270-f006:**
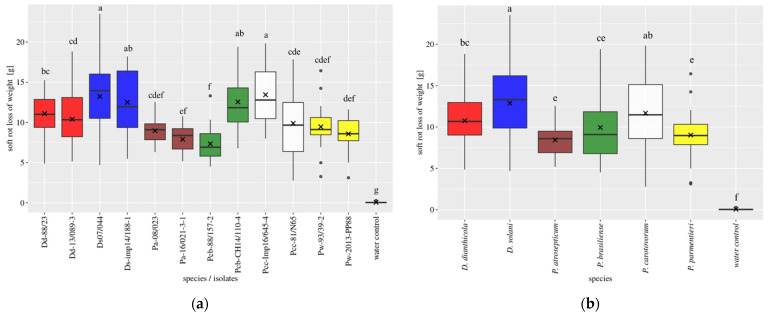
Soft rot loss of weight for in vitro potato slices infected with the different 12 isolates (**a**) or species (**b**) in the slices assay for 2020. Box plots of 20 repetitions for each isolate or 40 repetitions for each species; x indicates the mean; middle black line indicates the median. Different letters indicate statistical differences, according to pairwise multiple comparison (ANOVA based) of the mean value between the strains.

**Figure 7 microorganisms-09-02270-f007:**
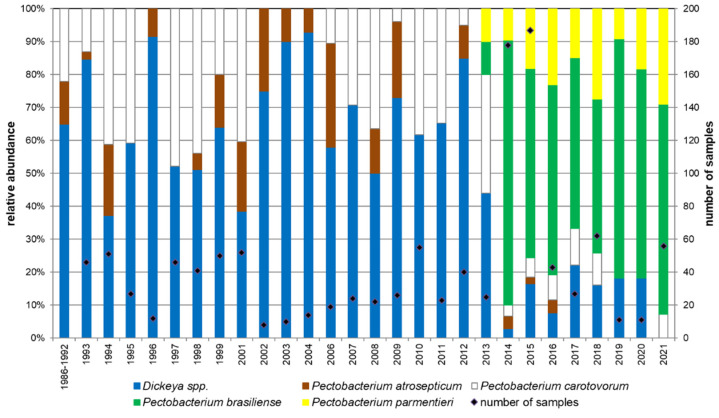
Relative abundance of different potato blackleg-inducing bacterial species sampled during the yearly Swiss survey from 1993 to 2021. Data until 2018 are already published elsewhere [11].

**Table 1 microorganisms-09-02270-t001:** Bacterial isolates used in this study, all from the Swiss Agroscope collection.

Genus	Species	Isolates	Year of Isolation
*Dickeya*	*dianthicola*	13/089-3	2013
	*dianthicola*	88/23	1988
*Dickeya*	*solani*	07/044	2007
	*solani*	14/188-1	2014
*Pectobacterium*	*atrosepticum*	08/023	2008
	*atrosepticum*	16/021-3-1	2016
*Pectobacterium*	*parmentieri*	93/39-2	1993
	*parmentieri*	2013-PP88	2013
*Pectobacterium*	*carotovorum*	Imp16/645-4	2016
	*carotovorum*	81/N65	1981
*Pectobacterium*	*brasiliense*	88/157-2	1988
	*brasiliense*	CH14/110-4	2014 ^1^

^1^ Determination of some species could have occurred many years after its isolation.

## Data Availability

The data that support the findings of this study are available from the corresponding author upon reasonable request.

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
