# Peer review of "Pathogenicity and Relative Abundance of Dickeya and Pectobacterium Species in Switzerland: An Epidemiological Dichotomy"

_microorganisms, 2021, doi:10.3390/microorganisms9112270_

Round 1

Reviewer 1 Report

The manuscript presents the study of the abundance and virulence of SRP, which were isolated in Switzerland. 

The work submitted for review was not prepared with due diligence, and the manuscript requires significant corrections before acceptance for publication.

The authors describe the results of a four-year study using Pectobacterium and Dickeya strains that have been isolated in Switzerland from symptomatic potatoes since 1986.
Two strains representing each of six bacterial species of pectinolytic bacteria were selected from a collection of strains systematically expanded in the last thirty-five years. However, it is not known on what basis the selection of strains was made? Judging by the information gathered in Table 1 they differ in the years of isolation. Was it the only selection criterium?
Maybe the observed differences between strains within a species are the result of selecting strains with extreme properties?
Without information on their properties, such as growth rate in in vitro conditions and the ability to acclimatize in the unfavorable environment conditions (like: growth at different temperatures, salinity, or water availability), it is challenging to draw unambiguous conclusions about the differences in pathogenicity between species. 

For testing virulence under laboratory conditions, a whole potato tuber test would better reflect the natural infection of plant tissues than is the case with potato tuber slices assay. 

The study also lacked data on the weather conditions prevailing in Switzerland in 2017 and 2018, in which greenhouse and field trials were carried out. Likewise, there is no information concerning years before 2014, when Dickeya was most frequently detected in Switzerland. Such data was provided only for 2020.
Weather data are readily available on the Internet and could support the hypothesis put forward in the discussion chapter.

      My objections are raised by the fact that the authors performed the greenhouse experiment only once. Why? There is no explanation rationalizing lack of replicate assays.

       What is more, the authors note in lines 201-202 the occurrence of disease symptoms on plants in the controls. However, no specific data was given on how many plants in control were symptomatic? 

Was it examined whether and what bacteria were present in the water used in the control sample?

What is more, there is no information on whether the presence of pectinolytic bacteria in the soil used for plant cultivation was checked?

Without such information, the conclusions drawn by the authors of the work do not have to be true.
Bacteria present in water or soil may support or inhibit the tested strains in the growth or induction of disease symptoms.

Furthermore, the greenhouse conditions allowing for the maintenance of a higher average daily temperature and higher humidity than in the field conditions were more conducive to the development of disease symptoms caused by Dickeya than Pectobacterium, especially P. atrosepticum, which has lower than other species optimal growth temp. 

Hence, in my opinion, the conclusions about the greater virulence of Dickeya than Pectobacterium in greenhouse conditions seem to be unfounded.

Recently several new species of Dickeya and Pectobacterium have been described. Most of them cannot be identified solely on the basis of PCR with species-specific primers that have been used in this study. So, was the classification of the 12 tested Swiss strains based only on the above-mentioned PCR method? 

Furthermore, primers used to detect Dickeya do not allow for identifying and differentiating individual species within this genus. How the D.solani and D. dianthicola strains were identified in case of strain selected for the study?

Minor remarks:

At the beginning of the introduction section, from line 31 to line 51, the species names of bacteria are not italicized.

The materials and Methods chapter should be modified. There is no clear scheme of the presented study.
The historical data of SRP monitoring performed in Switzerland since 1986 are given in detail; however, this work describes only the three years period from 2019 till 2021.
Unfortunately, such precision was not observed in the description of the pathogenicity assays. The timing of field trials was given but not for the greenhouses assay. The reader is confused when which tests were performed. Thus, the research scheme detailing time schedule and which assays where performed when would significantly facilitate the reception of work.

Reviewer 2 Report

More updated literatures needed for reasons of study during 2020-2021.

More updated literatures to discuss your results during 2019-2021 still needed

Insert conclusion part with more details 100-120 words

English should be improved; grammar need for enhancement in many sentence and paragraphs.

See attached PDF

Round 2

Reviewer 1 Report

There is still a need to change the nomenclature of Pectobacterium carotovorum strains in several places in the manuscript. In 2019, the subspecies P. c. subsp. carotovorum was elevated to the rank of a species (Portier et al. 2019), therefore the currently valid taxonomic position should be entered in lines 31, 107, 162, 446, and in Table 1